# Growth Optimisation and Kinetic Profiling of Diesel Biodegradation by a Cold-Adapted Microbial Consortium Isolated from Trinity Peninsula, Antarctica

**DOI:** 10.3390/biology10060493

**Published:** 2021-06-02

**Authors:** Ahmad Fareez Ahmad Roslee, Claudio Gomez-Fuentes, Nur Nadhirah Zakaria, Nor Azmi Shaharuddin, Azham Zulkharnain, Khalilah Abdul Khalil, Peter Convey, Siti Aqlima Ahmad

**Affiliations:** 1Department of Biochemistry, Faculty of Biotechnology and Biomolecular Sciences, Universiti Putra Malaysia, Serdang 43400, Malaysia; fareezlee@yahoo.com (A.F.A.R.); nadhirahairakaz@gmail.com (N.N.Z.); noorazmi@upm.edu.my (N.A.S.); 2Department of Chemical Engineering, Universidad de Magallanes, Avda. Bulnes, Punta Arenas 01855, Chile; claudio.gomez@umag.cl; 3Center for Research and Antarctic Environmental Monitoring (CIMAA), Universidad de Magallanes, Avda. Bulnes, Punta Arenas 01855, Chile; 4Department of Bioscience and Engineering, Shibaura Institute of Technology, College of Systems Engineering and Science, 307 Fukasaku, Minuma-ku, Saitama 337-8570, Japan; azham@shibaura-it.ac.jp; 5Faculty of Applied Sciences, School of Biology, Universiti Teknologi MARA, Shah Alam 40450, Malaysia; khali552@uitm.edu.my; 6British Antarctic Survey, NERC, High Cross, Madingley Road, Cambridge CB3 0ET, UK; pcon@bas.ac.uk; 7National Antarctic Research Centre, B303 Level 3, Block B, IPS Building, Universiti Malaya, Kuala Lumpur 50603, Malaysia

**Keywords:** diesel, microbial consortium, biodegradation, response-surface methodology (RSM), kinetic model

## Abstract

**Simple Summary:**

Diesel fuel is very crucial for anthropogenic activities in Antarctica and the surges in annual demand mean higher likelihood of spillages from improper handling during transportation, storage and disposal processes. The impacts can be very extensive or well-contained depending on the scale of the spills as well as the terrain involved. Nevertheless, the freezing temperature and prolonged solar irradiance in the south pole greatly hampered the natural attenuation and photovolatilisation of petrogenic hydrocarbons, contributing to their persistency. The most susceptible groups are the soil microorganisms, mosses, seabirds and pinnipeds as they are easily found near the shore where hydrocarbons spillage is very common. Microbial bioremediation is a well-established approach in restoring many hydrocarbons-polluted areas, thus the current study focused on the optimisation and application of locally isolated microbial consortium to simulate the in situ diesel clean-up process in aqueous medium. This study highlights the ability of the selected consortium to degrade diesel almost completely at moderately low temperature, suggesting its potential application in Antarctic settings.

**Abstract:**

Pollution associated with petrogenic hydrocarbons is increasing in Antarctica due to a combination of increasing human activity and the continent’s unforgiving environmental conditions. The current study focuses on the ability of a cold-adapted crude microbial consortium (BS24), isolated from soil on the north-west Antarctic Peninsula, to metabolise diesel fuel as the sole carbon source in a shake-flask setting. Factors expected to influence the efficiency of diesel biodegradation, namely temperature, initial diesel concentration, nitrogen source type and concentration, salinity and pH were studied. Consortium BS24 displayed optimal cell growth and diesel degradation activity at 1.0% NaCl, pH 7.5, 0.5 g/L NH_4_Cl and 2.0% *v*/*v* initial diesel concentration during one-factor-at-a-time (OFAT) analyses. The consortium was psychrotolerant based on the optimum growth temperature of 10‒15 °C. In conventionally optimised media, the highest total petroleum hydrocarbons (TPH) mineralisation was 85% over a 7-day incubation. Further optimisation of conditions predicted through statistical response-surface methodology (RSM) (1.0% NaCl, pH 7.25, 0.75 g/L NH_4_Cl, 12.5 °C and 1.75% *v*/*v* initial diesel concentration) boosted mineralisation to 95% over a 7-day incubation. A Tessier secondary model best described the growth pattern of BS24 in diesel-enriched medium, with maximum specific growth rate, *μ*_max_, substrate inhibition constant, K_i_ and half saturation constant, K_s_, being 0.9996 h^−1^, 1.356% *v*/*v* and 1.238% *v*/*v*, respectively. The data obtained suggest the potential of microbial consortia such as BS24 in bioremediation applications in low-temperature diesel-polluted soils.

## 1. Introduction

Antarctica is a continent that is isolated from major centres of human population and industry, one of the most pristine regions on Earth. Nonetheless, human contact with the continent, and in particular its ice-free coastal areas, primarily in the form of international polar research programmes and the lucrative ecotourism industry has led to a rapid increase in human activity compared to even only two to three decades ago [1,2,3]. Increased human activity has led to instances of anthropogenic environmental pollution and recognition of their potential negative impacts on the Antarctic environment and its biota.

Diesel and other hydrocarbon fuels, including their various chemical constituents, contribute a substantial proportion of hydrocarbon pollution events today as they are used extensively in power generation, vehicles and aircraft, and sometimes in waste incineration [4]. Although increasingly strict regulations applying to shipping have been adopted in recent years by the International Maritime Organisation, banning the use of heavy marine diesel, oil spills through marine accidents continue to occur. Lighter winter diesel variants such as Special Antarctic Blend (SAB) and marine gas oil are still permitted and extensively used by land-based stations in Antarctica, with average annual consumption amounting to around 2 million litres [5]. Fuel spills may occur during operations on land or snow/ice, ranging from small spills during vehicle refuelling to potentially major incidents during ship-to-station refuelling and leakage from poorly maintained pipelines or storage facilities [6].

Diesel fuels used in Antarctica typically have high vapour pressure due to their high kerosene content, but their volatility is considerably diminished at the low ambient temperatures typically experienced, exacerbated by the high surface albedo from snow cover that prevents photovolatilisation of hydrocarbons in the active soil layer, resulting in contaminants persisting in the environment for extended periods [7]. Soils in polar regions typically also have low capacity for natural attenuation owing to low biomass content and the extreme environmental conditions, resulting in the degradation or transformation of recalcitrant hydrocarbon compounds occurring much more slowly than in temperate and tropical regions [8,9].

In the marine environment, petrogenic hydrocarbons tend to bioaccumulate in lipid-rich tissues of aquatic organisms resulting in chronic impacts [10]. Extended exposure to polyaromatic hydrocarbons and their metabolites induces sub-lethal damage in multiple life stages, including reproductive impairment, behavioural alteration and immune and genetic toxicity in fish, seabirds, mammals and invertebrates [11]. Antarctic animals that spend time ashore, in particular the penguins and pinnipeds, are amongst the most susceptible to diesel pollution as they spend considerable time in coastal areas for feeding, mating and moulting and must periodically surface to breathe, hence exposing them to surface pollution [12].

Most decontamination practices currently applied to fuel spills in Antarctica require the use of specialised equipment and toxic dispersants, which are labour intensive, economically expensive and can themselves be detrimental to the environment [13,14]. As an alternative to these approaches, naturally‒occurring microbial groups with high degradation capacity for petrogenic hydrocarbons have potential to be exploited in bioremediation, with examples including bacterial genera such as *Rhodococcus* spp., *Burkholderia* spp., *Acinetobacter* spp., *Arthrobacter* spp. and *Pseudomonas* spp. [15,16,17]. The success of hydrocarbon bioremediation is dependent on a range of environmental stressors. Typical soil composition in Antarctica as well as the continent’s extreme environmental conditions are not favourable for microbial bioremediation [6]. The implementation of combined approaches including bioreactor-based remediation has become more viable due to their more rapid pollutant removal time and greater ability to control and maintain operational parameters, which is otherwise not feasible in open environment settings [18,19]. To achieve maximum efficiency of diesel degradation using microorganisms in laboratory or industrial settings, approaches that optimise parameters expected to have a large influence on growth and degradation rates, such as substrate type and availability, pH, temperature and salinity are often applied [20]. While superseded by response-surface methodology (RSM) in terms of resource and time requirement, the easy to apply conventional one-factor-at-a-time (OFAT) method continues to be used in the initial stages in order to provide approximate measurement ranges for each parameter [21]. The RSM approach uses complex statistical models to measure and interpret responses to multiple environmental factors and their pairwise interactions and also shortens the experimental runs required significantly [22,23].

The study of microbial growth patterns is also a focal point in predictive microbiology, which amalgamates mathematical modelling into microbiological principles [24]. The ability to predict growth patterns under the influence of various environmental factors is a key tool in appraising the behaviour of organisms of interest [25]. The classical model of sigmoidal microbial growth comprises, in sequence, a lag phase (adaptation), exponential phase (rapid replication), stationary phase (exhaustion) and death phase. In studies of microbial growth kinetics, a number of models have been developed to describe microbial growth patterns, with well-known examples including Monod, Haldane, Luong and Tessier models [16]. Growth is normally assessed as the numbers of colony forming units, sometimes also using optical density (OD) as an indirect measurement [26].

In this study, microbial consortia obtained from a total of 28 soil samples collected close to the Chilean Bernardo O’Higgins Station (Trinity Peninsula, north-west Antarctic Peninsula) were screened for diesel-degrading activity. The primary focus was to select one of the most effective consortia, then optimise its diesel biodegradation efficiency using both OFAT and RSM approaches. The statistically optimised conditions will provide a prior insight into which environmental parameters will need to be adjusted if the consortium is to be applied for diesel clean-up in Antarctic territories. The secondary element of the study was to incorporate kinetic modelling of microbial growth of selected consortium at different initial diesel concentrations using various secondary growth kinetics.

## 2. Materials and Methods

### 2.1. Sampling and Storage

A total of 28 soil samples were collected using sterile equipment within a 50-m radius of the Chilean Bernardo O’Higgins Research Station in January 2019 (Figure 1). Each sample comprised approximately 20 g soil obtained from the upper 15–20 cm of the soil profile, stored in sterile sealed collection tubes and kept at −20 °C on station and during return to Malaysia (approximately two months). Soil BS24 used throughout the study was obtained from an area previously identified to be contaminated with diesel hydrocarbons. The soil was oven-dried at 90 °C for 2 days or until constant mass was obtained, then segregated based on particle size using a rotary sifter RO-TAP RX-29-10 (W.S. Tyler, Mentor, OH, USA) with mesh size No. 8, No. 20, No. 70 and No. 100 following the ASTM D421-85 protocol [27]. Wet and dry mass was measured using weighing balance for soil profile determination, while soil TPH quantification followed the simplified gravimetric protocol as described in Villalobos et al. [28]. The soil pH, salinity and temperature were measured by using portable probes (LAQUAtwin Pocket Meters, Horiba Scientific, Kyoto, Japan).

### 2.2. Microbial Culture Medium

The carbon substrate (diesel fuel) used in this study was obtained from a local filling station (PETRONAS) in Selangor, Malaysia. Dehydrated nutrient broth (NB) used for consortium culturing in this study contained, per litre, 3.0 g beef extract and 5.0 g peptone. A standardised Bushnell‒Haas (BH) salt broth was used for the optimisation of growth conditions and diesel degradation [29]. Basal BH broth used in this study contained, per litre, 1.00 g K_2_HPO_4_, 1.00 g KH_2_PO_4_, 1.00 g NH_4_NO_3_, 0.20 g MgSO_4_, 0.05 g FeCl_3_ and 0.02 g CaCl_2_ in dH_2_O, with the addition of 20 g of bacteriological agar for solid media preparation. The pH of the BH media was adjusted to 7.0 ± 0.2 at room temperature by addition of HCl or NaOH prior to autoclaving at 121 °C for 15 min.

### 2.3. Consortium Isolation and Screening

A thawed sub-sample from each soil sample weighing 1.00 g was homogenised by vigorous shaking on vortex mixer for 15 min before subsequently inoculated into 10 mL of NB and incubated on an orbital shaker at 10 °C, 150 rpm for 2 d. The master culture was sub-divided into 500 μL starter cultures, with the addition of equal volume 50% glycerol stock for storage at −80 °C to ensure non-biased microbial composition between experimental sets. For subsequent use, the starter culture was thawed at room temperature before being aliquoted into 50 mL fresh NB and incubated using the aforementioned settings. Microbial pellets were then harvested by centrifugation (7000× *g*, 4 °C for 15 min), rinsed twice with 1× phosphate-buffered saline (PBS) (137 mM NaCl, 2.7 mM KCl in 10 mM phosphate buffer, pH 7.4) and adjusted to OD_600nm_ = 1.0 ± 0.1 prior to screening for diesel-degrading capacity in BH medium supplemented with 1% *v*/*v* sterilised diesel (filtered through a 0.45-μm PE membrane) [15]. Screening involved assessing the ability of each consortium to degrade diesel using a gravimetric method [28] and by measuring the OD_600nm_ of the broth, as described below.

### 2.4. Evaluation of Microbial Growth and Diesel Biodegradation

Evaluation of microbial growth was carried out spectrophotometrically by measuring OD_600nm_. A 1-mL aliquot of each culture was regularly taken at 24 ± 0.5 h intervals over a 7-day incubation and transferred into individual plastic cuvettes. The OD_600nm_ values of triplicates were measured immediately using a UV‒Vis spectrophotometer (Jenway 7305, Cole‒Parmer, IL, USA) and the mean calculated. Analysis of diesel biodegradation efficiency (BE) by each consortium was concurrently carried out following the incubation period using *n*-hexane extraction (1:1 medium to solvent ratio) with the conical flask immersed in 230 V ultrasonic water bath (XUBA 1, Grants Instruments Ltd., Royston, UK) for 10 min [28,30]. After 10 min extraction, the top mobile phase was transferred into a pre-weighed glass dish and concentrated under a fume hood. The glass dish was weighed a second time once the solvent had completely evaporated to obtain the final mass of diesel residue.

The gravimetric measurement of diesel mineralisation was expressed as the percentage efficiency of hydrocarbons mineralised (*BE*) relative to the abiotic loss of hydrocarbons in experimental controls (due to hydrocarbons volatilisation and aerosolisation) using the equation:$$BE\;\left(\%\right)=100-\left(W_{r}\times 100/W_{i}\right)$$
where *BE* (%) = percentage biodegradation efficiency, $W_{r}$ = residual mass of diesel in sample, $W_{i}$ = residual mass of diesel in control. The controls used consisted of only the BH medium and diesel at fixed concentration (except when testing the effects of diesel concentration).

### 2.5. Optimisation Using One-Factor-At-A-Time (OFAT) Approach

The consortium showing the most effective diesel degradation was selected for further study. Preliminary growth optimisation and assessment of diesel-degradative capability of consortium BS24 were carried out using the conventional OFAT approach based on six selected parameters: salinity (0.5, 1.0, 1.5, 2.0, 2.5, 3.0, 3.5% *w*/*v* NaCl), pH (4, 5, 6, 7, 8), type of inorganic nitrogen source (NaNO_2_, NaNO_3,_ NH_4_Cl, (NH_4_)_2_SO_4_, (NH_4_)_2_HPO_4_ or NH_4_NO_3_), most effective nitrogen source concentration (0.1, 0.2, 0.5, 1.0, 1.5, 2.0 g/L), temperature (10, 15, 20, 25, 30 °C) and initial diesel concentration (0.5, 1.0, 2.0, 3.0, 4.0% *v*/*v*). The effects of each subsequent parameter were investigated after fixing the previously considered parameters at their optimised values. The default growth settings were: 1% NaCl, pH 7, NH_4_NO_3_ as the nitrogen source, 1 g/L nitrogen source, temperature of 10 °C and 1% *v*/*v* initial concentration of diesel [15]. The culture media were prepared by inoculating 1 mL of adjusted cell suspensions (OD_600nm_ = 1.0 ± 0.1) into 250-mL conical flasks containing 50 mL BH broth supplemented with diesel (except for experimental controls). The flasks were incubated at 10 °C on an orbital shaker at 150 rpm for 7 d. All assays including controls were performed in triplicate. Evaluation of the effects of the individual parameters on growth and degradation followed the procedures described above.

### 2.6. Optimisation Using Response-Surface Methodology (RSM) Statistical Approach

Further studies on growth optimisation and assessment of diesel-degradative capability of consortium BS24 were carried out using the statistical approach of response-surface methodology (RSM). Typically, RSM comprises a combination of Plackett–Burman design (PBD) [31] and Box–Wilson central composite design (CCD) [32]. The parameters previously optimised during conventional OFAT were further screened via a PB factorial design to identify those that were significant. The experimental design was developed and analysed using the statistical software Design Expert 7.0 (Stat-Ease Inc., Minneapolis, MN, USA), within which each parameter (initial diesel concentration, nitrogen source concentration, pH, temperature and salinity) was evaluated at low and high levels (−1 and +1, respectively) (Table 1). The *BE* of TPH was analysed as the response variable in the identification of significant factors. CCD was then employed to construct the response surface of the identified significant parameters. The effects of each of these parameters on microbial growth and hydrocarbon biodegradation were analysed at two axial points (alpha), two factorial points and a single central point (+2/−2, +1/−1 and 0, respectively). The *BE* values of TPH were used as the response variable and fitted to a second-order polynomial regression model comprising linear, quadratic and interaction coefficients to predict the optimal conditions, thereby also identifying any significant interactions between the parameters.
$$Y=\beta_{0}+\sum_{i=1}^{k}\beta_{i}X_{i}+\sum_{i=1}^{k}\beta_{ii}X_{i^{2}}+\sum_{1<i<j}^{k}\beta_{ij}X_{i}X_{j}$$
where *Y* represents the predicted response; *X* the independent factors that significantly influence *Y*; *k* the number of factors; *β*_0_ a constant term; *β*_i_ the *i* th linear coefficient; *β*_ii_ the *i* th quadratic coefficient and *β*_ij_ the *ij* th interaction coefficient. The significance of each coefficient in the equation was determined by Fischer’s F-test and ANOVA (*p* < 0.05).

#### Model Validation

The resulting statistical models were validated for their reproducibility and robustness by performing an experiment using the predicted optima. The experimental values obtained were then compared against the predicted values using one-way analysis of variance followed (if significant) by a post hoc Tukey’s test (SPSS Inc., Armonk, NY, USA). In all analyses, *p* < 0.05 is accepted as significant.

### 2.7. Kinetics of Modelling Consortium BS24 Growth in Diesel Medium

The data required for mathematical modelling were obtained from the exponential phase of microbial growth as described above. In order to estimate the growth kinetic parameters of consortium BS24 in diesel-enriched medium, the maximum specific growth rate, *μ_max_*, achieved with each initial diesel concentration was calculated from the steepest slope. The input values for x- and y-axes were fitted to various non-linear regression models (Tessier, Monod, Haldane, Aiba, Edwards, Yano, Luong) [33,34,35,36,37,38,39] by enabling the auto-initial guesses function in CurveExpert Professional software v.7 (Chattanooga, TN, USA).

#### Statistical Analyses for Kinetics Modelling

Common statistical measurements such as the root-mean square error (RMSE), coefficient of determination, *R*^2^, adjusted-*R*^2^, corrected Akaike Information Criterion (AICc), accuracy factor (A_*f*_) and bias factor (B_*f*_) are used to assess the robustness and the presence of significant differences between models. The RMSE measures the extent of residual spreading relative to the regression line and has a direct correlation to the coefficient of determination such that the value of RMSE is zero when the *R*^2^ value equals one (unity) [40]. While an *R*^2^ value approaching unity reflects a good model fit, the implementation of different models in non-linear regression analyses typically accounts for distinct sets of parameters, thus this value alone is insufficient to compare data fitness between models, which entails additional calculation of adjusted-*R*^2^, AICc, A_*f*_ and B_*f*_ [22,26].

## 3. Results

### 3.1. Consortium Isolation and Screening

After a 7-day incubation at 15 °C in Bushnell–Haas broth (BH) supplemented with 1% *v*/*v* initial diesel concentration, the cell growth and percentage biodegradation (BE) of TPH by 28 consortia were obtained, with all cultured consortia being able to assimilate diesel hydrocarbons to a certain extent except for BS17 (Figure 2). From the original 28 samples, six consortia displaying ≥80% *BE* were subjected to a secondary screening with 2% *v*/*v* initial diesel concentration leading to consortium BS24 being chosen for further optimisation studies based on its high growth and TPH degradation as verified through one-way ANOVA and a post hoc Tukey’s test (Figure 3) (Appendix A).

### 3.2. Soil BS24 Characterisation

The soil is comprised of, per kg, 79.85% gravel, 6.00% sand, 2.90% silt, 2.00% clay, 8.59% moisture, and 0.66% (approximately 5.61 g) weathered diesel. Simple biochemical assessment of the soil showed that it is near-neutral at pH 6.81, shows salinity of 0.9 ppt and an average ground temperature of 4 °C at the time of collection.

### 3.3. Optimisation of Growth Medium Using OFAT Approach

Culture conditions were initially optimised using the OFAT approach, with selection of six parameters expected to influence microbial growth and diesel degradation: salinity, pH, temperature, type and concentration of nitrogen source, and initial diesel concentration. Figure 4a shows the effect of NaCl concentrations on microbial growth and degradation capacity, both being highest in the range of 1.0‒1.5% *w*/*v*. A significant reduction in both responses was noted when NaCl concentration was increased to 3.5% *w*/*v*. The pH optimum of consortium BS24 for both growth and TPH degradation was pH 7.5‒8 (Figure 4b). The use of NH_4_Cl as nitrogen source resulted in significantly increased microbial growth and TPH degradation than the other inorganic sources trialled (Figure 4c). Subsequent trials using different NH_4_Cl concentrations ranging from 0.1 to 2.0 g/L confirmed that, while BS24 was able to grow at all concentrations used in the trials, the final degradation percentage declined markedly beyond 0.5 g/L NH_4_Cl (Figure 4d). Both growth and degradation rates reached maximum at 10 °C, at which 82.24% TPH mineralisation was achieved, then declined steadily as the temperature was increased up to 25 °C (Figure 4e). Finally, consortium BS24 showed no significant differences in growth or diesel biodegradation in media supplemented with initial diesel concentrations of 0.5‒2.5% *v*/*v*, while both responses consistently decreased at higher concentrations (Figure 4f).

### 3.4. Optimisation of Growth Using RSM Approach

In this study, three-dimensional response-surface prediction was achieved through two-part analyses employing the PBD and the CCD. A two-level PBD experiment was initially performed in triplicate to reduce the experimental runs required, using the minimum and maximum values (coded as −1 and +1, respectively), where the values for each parameter were gauged from the previous ANOVA-validated OFAT analyses using SPSS software. In this phase, substantial differences were achieved in TPH degradation, between 34.43% (run 1) and 91.23% (run 12) (Table 2).

Following the PBD analysis, four significant factors A (*p* = 0.0160), B (*p* = 0.0010), D (*p* = 0.0224), and E (*p* = 0.0018) representing pH, temperature, NH_4_Cl concentration and diesel concentration, respectively, were selected for further optimisation in CCD analysis (Table 3). The remaining non-significant factor C (*p* = 0.5297) was not further optimised during the CCD analysis, and was only employed at its low (−) or high (+) levels. Generally, it is recommended that the (+) value be employed when the factor exerted a positive influence and the (−) value when a negative influence was exerted. The reliability of experimental data was confirmed through the coefficient of determination, *R*^2^, with a value of 0.9992, denoting a very good fit and strong correlation between the experimental and predicted values (Table 3). The resulting significant *p*-value of 0.0453 indicated that the generated model was statistically significant with the equation for diesel degradation (Y) being:Y = + 80.1625 + 22.51667A − 4.15033B − 6.87677D − 55.25000E

The four significant factors were then further optimised via five-level CCD analysis consisting of 27 experimental runs with varying permutations as generated by the Design-Expert 7 software algorithms (Stat-Ease, Minneapolis, MN, USA). Table 4 shows the average experimental TPH degradation efficiency in response to the growth culture conditions, with a lowest value of 17.22% (run 1) and a highest degradation value of 95.49% (run 15).

The reliability of the CCD model was confirmed by the high value of the coefficient of determination (*R*^2^ = 0.9758) and identified the significant variables and interaction terms (Table 5). The analysis generated the following equation representing Log_10_ TPH degradation (Y):Y = + 80.13+ 7.44B − 3.92AB − 5.08AD − 8.13AE − 5.19A^2^ − 16.03B^2^

Three response-surface contour plots (Figure 5) were generated based on the significant interaction terms identified in Table 5. The factors involved were pH, temperature, NH_4_Cl concentration and initial diesel concentration, denoted as A, B, D and E, respectively. Figure 5a shows the response resulting from the interaction between temperature and salinity while maintaining the D and E values at 0.75 g/L and 2.25% *v*/*v*, respectively. The contour plot indicated the highest degradation efficiency of 94.77% at 12.5 °C and pH 7.25. Figure 5b depicts the response resulting from the interaction between temperature and NH_4_Cl concentration while maintaining A and E at constant pH 7.25 and 2.25% *v*/*v*, respectively. The highest degradation efficiency of 94.77% was recorded at 12.5 °C and 0.75 g/L NH_4_Cl. Lastly, Figure 5c shows the plot resulting from the interaction between temperature and initial diesel concentration while maintaining A and D at constant pH 7.25 and 0.75 g/L, respectively. The highest degradation efficiency of 95.49% was recorded at 12.5 °C and 1.75% *v*/*v* diesel. The robustness of these models was validated by comparing the predicted response in CCD against the experimental runs. From statistical prediction, 87.84% from 2.0% *v*/*v* initial diesel would be degraded under optimised conditions of 13.75 °C, 0.7 g/L NH_4_Cl and pH 7.2. The post hoc Tukey’s test comparing with the experimental triplicates showed no significant difference between the predicted and observed values (*p* = 0.0913) and a correlation coefficient, R^2^ of 0.9789.

### 3.5. Growth Kinetic Modelling

In this experiment, the growth kinetic parameters of consortium BS24 in the presence of diesel were measured using the experimental specific growth rate, *μ*, for each initial diesel concentration, varying from 0.5 to 4.0% *v*/*v*. Figure 6 illustrates that the experimental *μ* values that conform to the typical substrate-inhibition model, initially displaying increasing trends before decreasing rapidly as the initial concentration of diesel continued to increase, in this case after 1.5–2.0% *v*/*v*. The maximum apparent specific growth rate, *μ**, was 0.0382 ± 0.0026 h^−1^, and occurred at 72 h incubation and 2.0% *v*/*v* initial diesel concentration. For comparative purposes, the specific growth rates of consortium BS24 were fitted to six non-linear regression models, Luong, Aiba-Edwards, Yano and Koga, Tessier, Haldane and Monod.

The mathematically best-fitting model describing the growth of consortium BS24 in diesel medium was the Tessier model displaying the overall highest value of *R*^2^ and adjusted-*R*^2^, low RMSE and AICc values along with the A*_f_* and B*_f_* values approaching unity (Table 6). However, other than the Monod model, all models’ tests provided a visually very similar fit to the experimental data across most of the range of initial diesel concentrations tested.

The biological coefficients calculated using the Tessier model, the maximum specific growth rate, *μ*_max_, substrate inhibition constant, K_i_, and half saturation constant, K_s_, were 0.9996 h^−1^, 1.356% *v*/*v* and 1.238% *v*/*v*, respectively. However, the value of *μ*_max_ obtained from curve interpolation represents only the hypothetical limit, while the true *μ*_max_ value occurred at a diesel concentration, S_m_, of 1.296% *v*/*v* with a value of approximately 0.0336 h^−1^. Therefore, the expression for the model was:$$\mu =0.0336\left[1-exp\left(\frac{S}{1.356_{i}}\right)-\exp\left(\frac{S}{1.238}\right)\right]$$

## 4. Discussion

In terrestrial habitats, hydrocarbonoclastic soil bacteria from diverse genera such as *Rhodococcus*, *Arthrobacteria*, *Pseudomonas*, *Acinetobacter* and *Sphingomonas* work synergistically to degrade various hydrocarbon contaminants, producing energy and biomolecules essential for growth through a series of complex catabolism processes [17,23,41]. However, at higher concentrations, diesel components such as PAHs and other lipophilic derivatives are potentially cytotoxic and able to weaken the integrity of the cell membrane due to their solvent effects [42].

A thorough understanding of the biogeochemical properties of a hydrocarbon-contaminated site is a prerequisite for the success of any attempt at microbial bioremediation. For instance, salinity is an environmental stressor known to significantly impede microbial propagation at high concentrations, a factor that is likely to be relevant in Antarctica’s ice-free coastal environments as well as in regions such as the McMurdo Dry Valleys where high salinities occur as a result of long-term accumulation of salts from geological weathering and infrequent precipitation in the absence of leaching and outwash [43]. While NaCl is a necessary component for normal membrane function and cellular activity, hypersaline environments, which are common in Antarctica, can cause excessive osmotic stress across the microbial cell membrane, disrupting the function of metabolic enzymes and potentially leading to extreme dehydration of cells [44]. In this study, consortium BS24 obtained from a coastal soil was subjected to exposure to NaCl concentrations range from 0.5 to 3.5%. The consortium showed optimal performance at NaCl concentration of 1‒1.5%, with performance dropping considerably at concentrations of 3% and greater, suggesting limited likelihood of its use in bioremediation application at coastal sites with strong marine influence.

Other abiotic factors such as pH also have an important influence in planning of bioremediation approaches. Most microbes perform best at near-neutral pH, although there are exceptions such as various Archaea that can tolerate extreme acidity or alkalinity. Tolerance of pH stress is often facilitated by physiological modification of the cell membrane to assist regulation of the intracellular pH level. Foong et al. [45] reported that pH in typical Antarctic soils varies from pH 6 to 9 depending on the minerals present and coastal proximity, with inland soils tending to be more alkaline. Soil recently polluted by diesel can have pH as low as 5.5. In a shake flask setting, pH changes are largely due to the accumulation of metabolic wastes, which can be compensated for through the use of suitable buffer systems. TPH degradation and microbial growth were both maximal in the current study between pH 7 and 7.5, consistent with similar studies [15,41,46].

The availability of appropriate nitrogen sources is also crucial in biodegradation processes, with various nitrogen-containing molecules being incorporated into the products of biodegradation as well as in the enzymes and co-factors in the metabolic processes involved in hydrocarbon biodegradation [47,48]. While access to an appropriate nitrogen source is crucial, concentration is also important [49]. Bokhorst et al. [50] mapped nitrogen concentration footprints in Antarctic soils in a study of the influence of marine vertebrate fertilisation of terrestrial ecosystems at a number of locations in the maritime Antarctic. Similarly, Lachacz et al. [51] documented the relationship between marine vertebrate guano sources and concentrations of NH_4_^+^ and NO_3_^−^ in coastal soils (8.86 g/kg soil and 2.79 g/kg soil, respectively), much greater than in inland soils not influenced by vertebrates.

Environmental temperature is perhaps the biggest challenge facing proponents of bioremediation in Antarctica both in terms of chronically low average temperatures close to freezing point and considerable fluctuation [20]. Optimum conditions as experimentally assessed are rarely relevant in the natural environment, resulting in poor outcomes. However, in studies such as that carried out here, important information can be derived from the calculated performance curves and response surfaces relating to performance at suboptimal levels of the identified influential variables; for instance, TPH may remain above 50% of the optimum level even at sub-optimal temperatures some distance from the measured optimum. In this study, consortium BS24 achieved optimum degradation at temperatures of 10‒15 °C, similar to that reported in previous studies of hydrocarbonoclastic psychrotolerant bacteria [41,52]. Maritime Antarctic soil temperatures do reach such temperatures for periods during the austral summer and spend considerable time in the lower suboptimal temperature range of 5–10 °C [53].

RSM is a well-established, high resolution approach with numerous potential applications both industrially as well as for scientific studies. As recognised elsewhere, RSM gives advantages over conventional OFAT, in particular producing precise estimation of influential factor interaction parameters with relatively less cost and time, effectively shortens experimental runs [21,54]. However, some of the widely acknowledged limitations of RSM in many studies are that any experiment with too many factors is often prone to complex model interpretation or sometimes may become totally inexplicable with just first order and second order polynomials [32]. A further disadvantage of RSM is poor outcomes prediction for a system beyond the range of study under consideration; therefore, it strongly relies on previous knowledge of the subject [21]. Until recently, only a handful of studies have attempted to model psychrotolerant microbial consortia through RSM. In the RSM-optimised culture examined here, the TPH degradation efficiency was boosted to 95% over the 85% achieved under OFAT. Further studies using different oil blends would be required in order to assess the performance of consortium BS24 when exposed to the specific diesel fuel types that are used in Antarctica, which will differ in composition details from the PETRONAS diesel blend used in this study.

While consensus in model choice has not been achieved, the aim is to apply the model that best describes the parameters from the experimentally generated growth curve. Comparative studies of different models typically measure the model fitness by calculating the bias factor (B*_f_*) and accuracy indices (A*_f_*) [16], coefficient of determination (*R*^2^), residual mean square error (RMSE) or the F-test [55], while studies also emphasise the use of direct comparisons of various model-predicted specific growth parameters [56,57]. The implementation of OD measurement to assess microbial growth response can have limited utility and in particular cannot be applied in highly turbid media or when solid growth matrices are used. The aforementioned problems are more commonly encountered in studies using fast-growing microbes typically attributed to temperate regions. Additionally, the use of biosurfactant-producing organisms may significantly affect both the OD measurement and extractable TPH due to the formation of micelles (dispersed oil droplets); otherwise, the less dense diesel usually floats on top of the aqueous medium throughout the incubation period. Despite these drawbacks, the advantages of using OD as a proxy for microbial growth include rapid quantification, relative simplicity and being non-destructive and cost effective when compared to other available techniques [58]. OD assessment also has utility in qualitative comparisons of growth of different microbial cultures or of the same culture under different conditions.

Low-weight hydrocarbons are prone to volatilisation and aerosolisation when shaken for a prolonged time, especially at elevated temperature set-ups to which it can contribute to erroneous analysis. It is therefore crucial to consider the abiotic loss in the calculation of TPH mineralisation, as seen in all bar figures in this study. As mentioned elsewhere, diesel is a myriad of hydrocarbons in which some of them were known to interfere with membrane integrity and function. In Figure 4f, the authors reported an intriguing interaction between the effects of increasing hydrocarbons concentration towards the microbial growths. The similar growth peaks (*p* < 0.05) at day 7 for diesel concentrations of 0.5% to 2.5% may be attributed to a form of microbial adaptation strategy which prefers individual cell survivability instead of actively undergoing cellular proliferation. At 3% and 4% of diesel concentrations, the microbial growth declined markedly as the cytotoxic effects become prominent and intolerable by the majority of the microbial population. However, this claim is crudely speculative and therefore needs further investigation.

## 5. Conclusions

In summary, this study demonstrated that the Antarctic soil microbial consortium BS24 showed optimum performance in degrading diesel hydrocarbons as its sole energy source at the moderately low temperature of 10‒15 °C. The use of OFAT and RSM approaches identified that the factors of temperature, pH, NH_4_Cl concentration, initial diesel concentration and some of their two-way interactions had significant influence on diesel degradation. The predicted RSM-optimised conditions for TPH biodegradation were a temperature of 12.5 °C, pH 7.25, 1.0% NaCl, 0.75 g/L NH_4_Cl and 1.75% diesel. However, we also highlight that the application of optimised conditions for in situ bioremediation in polar settings or other extreme environments is unlikely to be achieved as suboptimal conditions will inevitably predominate, meaning that understanding the shape of performance curves and response surfaces is also an important component in assessing levels of consortium performance under such conditions. In some circumstances, for instance if more drastic disturbance of soil structure is permitted after smaller pollution events, the use of more controlled systems such as bioreactors and indoor biostimulation may be appropriate.

## Figures and Tables

**Figure 1 biology-10-00493-f001:**
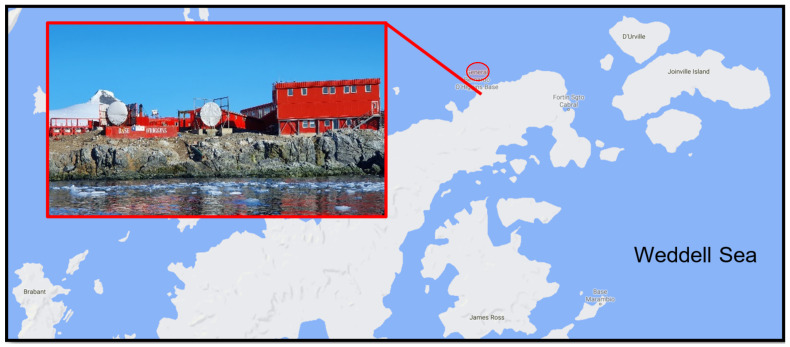
The Chilean year-round operating General Bernardo O’Higgins Riquelme Station located on the Trinity Peninsula, north-west Antarctic Peninsula (63°19′ S 57°54′ W).

**Figure 2 biology-10-00493-f002:**
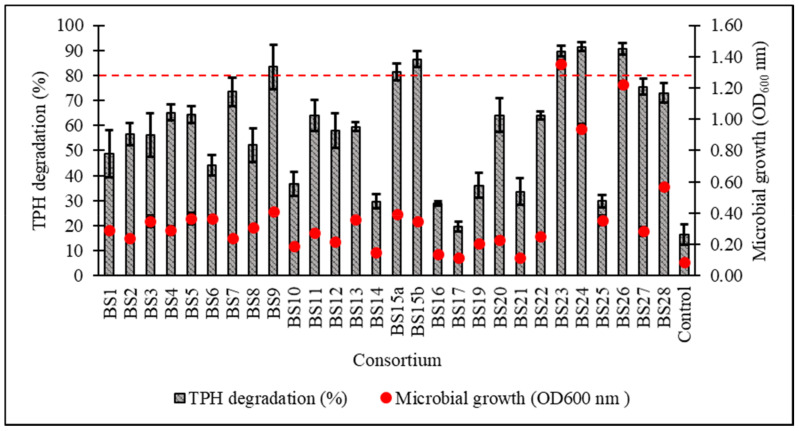
Percentage degradation of total petroleum hydrocarbons (TPH) and microbial growth (as OD_600_) after 7-day incubation in standardised Bushnell‒Haas (BH) broth enriched with 1% *v*/*v* initial diesel concentration. The dotted red line indicates ≥80% TPH degradation capacity. Error bars represent the mean ± standard deviation for experimental triplicates.

**Figure 3 biology-10-00493-f003:**
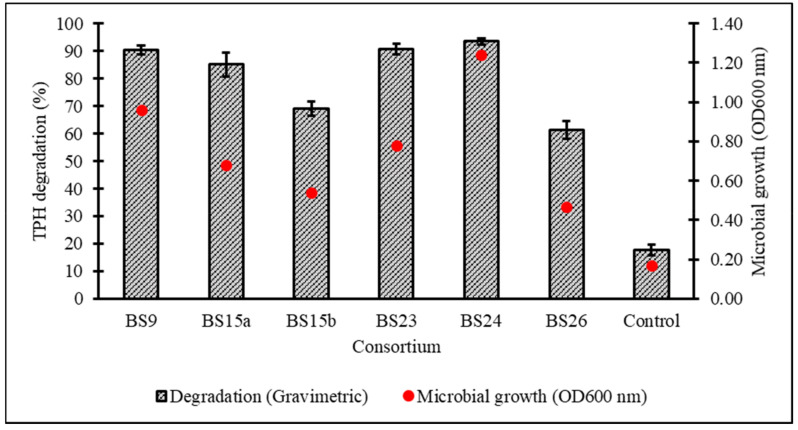
Percentage degradation of TPH and microbial growth (as OD_600nm_) achieved in secondary screening of six selected consortia in standardised BH broth enriched with 2% *v*/*v* initial diesel concentration. Error bars represent the mean ± standard deviation for experimental triplicates.

**Figure 4 biology-10-00493-f004:**
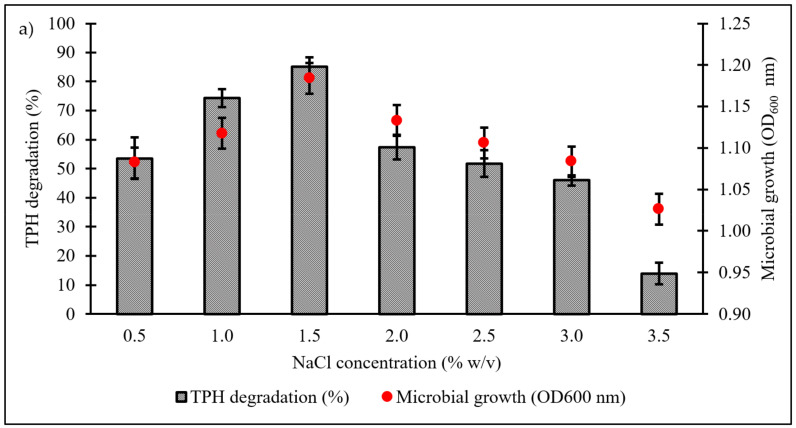
The effects of variation in levels of (**a**) NaCl concentration (% *w*/*v*); (**b**) pH; (**c**) inorganic nitrogen source; (**d**) NH_4_Cl concentration (g/L); (**e**) temperature (°C); and (**f**) initial diesel concentration (% *v*/*v*) on TPH degradation and microbial growth of consortium BS24 in 7-day incubation. Error bars represent the mean ± standard deviation for experimental triplicates.

**Figure 5 biology-10-00493-f005:**
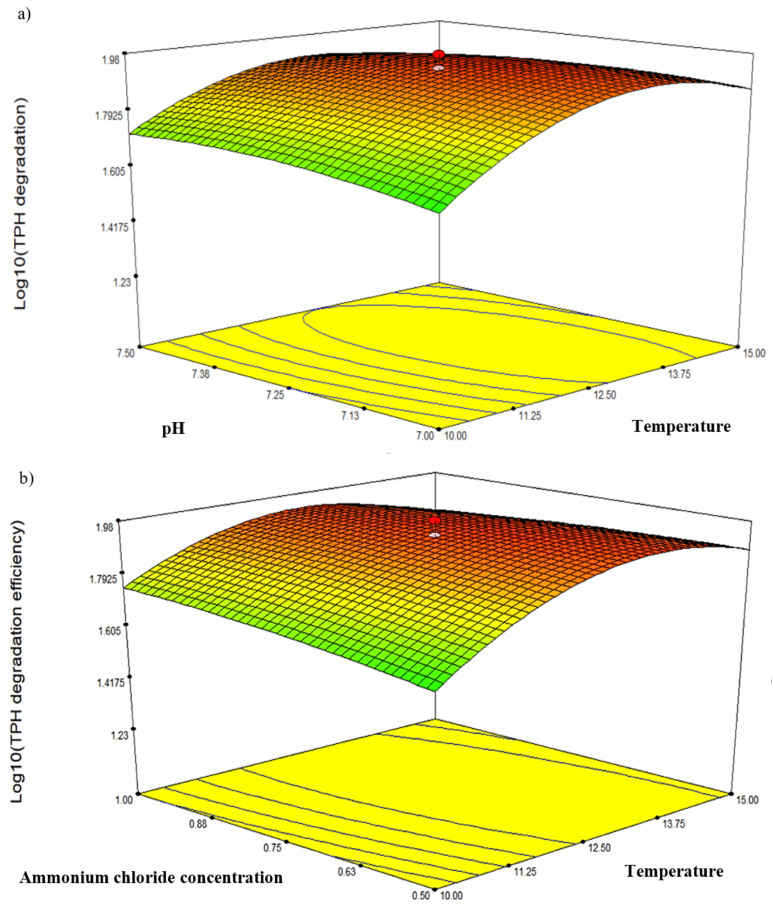
The three-dimensional response-surface plots showing the significant pairwise interactions between factors and the biodegradation efficiency by microbial consortium BS24. The contours (**a**–**c**) depict the interactions between pH and temperature, NH_4_Cl and temperature, and diesel concentration and temperature, respectively.

**Figure 6 biology-10-00493-f006:**
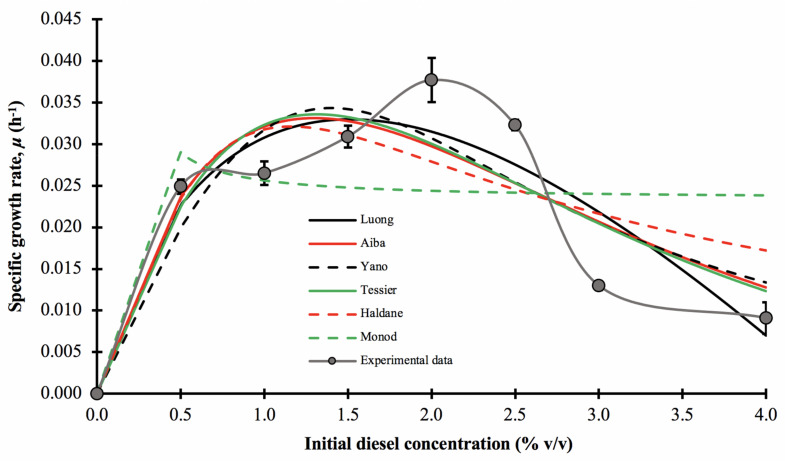
The effect of initial diesel concentration on the specific growth rate, *μ*, of consortium BS24, using OD_600nm_ as a proxy for microbial growth and the experimentally assessed and predicted specific growth rates fitted to selected mathematical models at different initial diesel concentrations. The error bars represent the mean ± standard deviation of experimental triplicates.

**Table 1 biology-10-00493-t001:** Factors employed for response-surface methodology (RSM) analysis and their respective levels.

Actual Factor	Coded Factor	Unit	Low Level (−1)	High Level (+1)
pH	A	−	7.0 ^a^	7.5 ^b^
Temperature	B	°C	10 ^c^	15 ^d^
Salinity	C	%	1.0 ^e^	1.5 ^f^
NH_4_Cl concentration	D	g/L	0.5 ^g^	1.0 ^h^
Diesel concentration	E	% *v/v*	2.0 ^i^	2.5 ^j^

Standard deviation for low and high levels of each factor: ^a^ ±7.108; ^b^ ±7.408; ^c^ ±0.720; ^d^ ±0.733; ^e^ ±3.081; ^f^ ±3.275; ^g^ ±4.586; ^h^ ±3.241; ^i^ ±3.976; ^j^ ±2.700.

**Table 2 biology-10-00493-t002:** Plackett–Burman design (PBD) permutation matrix for the identification of significant factors and the corresponding TPH degradation efficiency by consortium BS24.

Run	Experimental Value					TPH Degradation (%)
	A	B	C	D	E	
1	+	+	-	+	+	34.43
2	-	+	+	-	+	44.44
3	+	+	-	+	-	88.26
4	-	+	-	-	-	52.46
5	-	+	+	+	-	44.75
6	+	-	+	+	-	88.10
7	-	-	+	+	+	55.22
8	-	-	-	+	+	51.28
9	-	-	-	-	-	90.43
10	+	-	-	-	+	58.35
11	+	+	+	-	+	45.76
12	+	-	+	-	-	91.23

See Table 1 for factor identity (A−E). Degradation efficiency is represented by the mean ± standard deviation for experimental triplicates.

**Table 3 biology-10-00493-t003:** Analysis of variance of TPH degradation by consortium BS24 in PBD analysis.

Source	Sum of Squares	Df	Mean Square	F Value	*p*-Value
Model	4926.72	9	492.67	294.16	0.0453 *
A	115.12	1	115.12	61.18	0.0160 *
B	1927.99	1	1927.99	1024.63	0.0010 ***
C	12.05	1	12.05	0.57	0.5297
D	81.05	1	81.05	43.07	0.0224 *
E	1037.55	1	1037.55	551.40	0.0018 **
Residual	3.76	2	1.88		
Cor Total	4928.39	11			
Std. Dev.	1.37		Coeff. Determination, *R*^2^		0.9992
Mean	62.06		Adjusted-*R*^2^		0.9958
C.V. %	2.21		Predicted-*R*^2^		0.9889
PRESS	54.53		Adequate Precision		46.089

* *p* < 0.05, ** *p* < 0.01, *** *p* < 0.001.

**Table 4 biology-10-00493-t004:** Central composite design (CCD) permutation matrix for the screening of significant factors and the corresponding TPH degradation efficiency by consortium BS24.

Run	Experimental Value				TPH Degradation (%)
	A	B	D	E	
1	0	−2	0	0	17.22
2	−1	−1	−1	−1	26.14
3	+1	−1	−1	−1	37.41
4	−1	−1	+1	−1	37.37
5	+1	−1	+1	−1	43.83
6	−1	−1	−1	+1	45.52
7	+1	−1	−1	+1	51.53
8	−1	−1	+1	+1	58.08
9	+1	−1	+1	+1	62.19
10	−2	0	0	0	68.79
11	+2	0	0	0	67.53
12	0	0	−2	0	74.43
13	0	0	+2	0	81.61
14	0	0	0	−2	86.10
15	0	0	0	+2	95.49
16	0	0	0	+1	94.77
17	0	0	0	+1	84.36
18	0	0	0	+1	79.23
19	−1	+1	−1	−1	80.33
20	+1	+1	−1	−1	69.05
21	−1	+1	+1	−1	70.98
22	+1	+1	+1	−1	63.52
23	−1	+1	−1	+1	62.58
24	+1	+1	−1	+1	63.41
25	−1	+1	+1	+1	58.69
26	+1	+1	+1	+1	41.77
27	0	+2	0	0	32.36

See Table 1 for factor identity (A−E). Degradation efficiency is represented by mean ± standard deviation of experimental triplicates.

**Table 5 biology-10-00493-t005:** Analysis of variance of TPH degradation by consortium BS24 in experimental CCD.

Source	Sum of Squares	Df	Mean Square	F Value	*p*-Value
Model	0.81	14	0.058	34.56	<0.0001 ***
A	5.55 × 10^−5^	1	5.55 × 10^−5^	0.033	0.8586
B	0.14	1	0.14	81.37	<0.0001 ***
D	1.54 × 10^−3^	1	1.54 × 10^−3^	0.92	0.3546
E	6.56 × 10^−3^	1	6.56 × 10^−3^	3.91	0.0664
AB	0.02	1	0.02	11.88	0.0047 **
AD	0.032	1	0.032	18.83	0.1581
AE	0.08	1	0.08	47.9	0.3213
BD	3.80 × 10^−3^	1	3.80 × 10^−3^	2.27	0.0001 ***
BE	1.79 × 10^−3^	1	1.79 × 10^−3^	1.07	<0.0001 ***
DE	1.52 × 10^−3^	1	1.52 × 10^−3^	0.9	0.3605
A^2^	0.022	1	0.022	13.50	0.0032 **
B^2^	0.46	1	0.46	279.50	<0.0001 ***
D^2^	6.816 × 10^−3^	1	6.816 × 10^−3^	4.10	0.0657
E^2^	4.432 × 10^−5^	1	4.432 × 10^−5^	0.027	0.8730
Residual	0.02	12	1.68 × 10^−3^		
Lack of Fit	0.017	10	1.67 × 10^−3^	0.98	0.6042
Pure Error	3.40 × 10^−3^	2	1.70 × 10^−3^		
Cor Total	0.83	26			
Std. Dev.	0.041		Coeff. Determination, *R*^2^		0.9760
Mean	1.76		Adjusted-*R*^2^		0.9480
C.V. %	2.32		Predicted-*R*^2^		0.8750
PRESS	0.10		Adequate Precision		25.282

See Table 1 for factor identity (A-E). ** *p* < 0.01, *** *p* < 0.001.

**Table 6 biology-10-00493-t006:** Statistical analysis of non-linear models applied for estimation of growth kinetics using consortium BS24.

Model	*R* ^2^	*n*	Adj-*R*^2^	RMSE	AICc	A*_f_*	B*_f_*
Monod [34]	0.0254	2	−0.1695	0.0112	−62.388421	1.000	1.000
Aiba [36]	0.7906	3	0.7402	0.0067	−65.53920	1.002	1.002
Tessier [33]	0.8559	3	0.7479	0.0066	−65.60998	1.002	1.002
Haldane [35]	0.7258	3	0.6162	0.0081	−62.80734	1.002	1.002
Yano [38]	0.8144	4	0.6564	0.0076	−58.14494	1.001	1.001
Luong [39]	0.8145	4	0.7402	0.0066	−60.31196	1.000	1.000

## Data Availability

Not applicable.

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
