# Peer review of "Growth Optimisation and Kinetic Profiling of Diesel Biodegradation by a Cold-Adapted Microbial Consortium Isolated from Trinity Peninsula, Antarctica"

_biology, 2021, doi:10.3390/biology10060493_

Round 1

Reviewer 1 Report

The manuscript by Roslee et al. describes an investigation into the diesel-degradation capabilities of microbial communities in Antarctic soil samples. After testing the efficiency of diesel degradation in different soil samples, the authors picked the most efficient sample and continued their analysis using one-factor-at-a-time and statistical response surface methodology analyses. This resulted in the identification of several factors that significantly influenced growth and diesel degradation efficiency. The study is in large parts well-conducted; however, some aspects need clarification before a potential publication in Biology. One major shortcoming of the study is that the authors do not analyze the taxonomic composition of their most promising consortium, which limits the applicability of this study even beyond the adverse geological and physicochemical conditions in the Antarctic mentioned by the authors.

Throughout the text:

Please provide manuscript line numbers (e.g. use the manuscript template files you can find on the Biology Journal website).

Introduction:

The introduction is well-written and provides sufficient background of the topic. However, it is very long. The authors could easily shorten the introduction by removing some details or moving them to the discussion, e.g. the detailed description of model fitness calculations, or of the use and shortcomings of OD measurements to estimate growth.

References:

Please use original research article only to support any scientific claims. E.g. reference 5 in the first paragraph does not seem to be a proper reference for the claim that the annual Diesel consumption amounts to 2 million liters annually. Please remove (or move) this reference and add a reference that actually provides data for this number. Similarly, references are missing in other instances, e.g. for the claims in the last sentence of the first paragraph.

Material and methods:

The materials and methods section is generally well-written and provides a lot of background information, which helps readers who are not from the field to better understand some of the more complex methods.

Paragraph 2.1: Given that 20 g of soil sample were collected and later used in single aliquots in different experiments the phrase “consortium” used throughout the manuscript seems incorrect. If the soil samples were not homogenized before freezing one would expect many different “consortia” to occur in 20 g of soil, which would later be individually sampled for the following experiments. Please explain if the samples were well mixed before freezing or not.

Paragraph 2.3: It does not become clear how the diesel degrading cultures were inoculated. What happened after adjusting the OD to 1.0? Were aliquots of this used to inoculate diesel-containing medium? What was the start OD? Were diesel containing cultures incubated statically or were they shaken? What type of incubation vessels were used? (see paragraph 2.5)

Paragraph 2.4: After describing the shortcomings of using OD to estimate growth in cultures with substrates that influence turbidity in the introduction, it seems counterintuitive that the authors chose this method for their analysis. At a minimum, the authors should describe how diesel influenced OD measurements throughout their incubations and discuss possible effects on the outcome of their analysis. It might also help readers who are not familiar with this kind of experiments to provide more details on the cultures, e.g. by describing whether the diesel was floating on top of the aqueous medium or whether it was dispersed.

Results:

The results are generally well-written, however the authors decided not to show selected results (e.g. 7 days growth curves) for some experiments. Since Biology requires that authors publish all experimental controls and make full datasets available, please provide these data in the supplement.

Paragraph 3.1: The authors describe in the Material and Methods section that they sampled the consortia-screening cultures every 24 h for OD measurement but do not show any data on this. Their results would be more convincing if these growth curves would be provided in a supplementary file and the reader could follow biomass increase in a time-dependent manner. This could also help to clarify the discrepancies in biomass yield between the first and second round of screening: One could roughly expect twice the biomass being produced when the amount of carbon source is doubled (1% vs. 2% diesel) given that the carbon source is quantitatively used up under both settings. However, this is not the case in the presented results. On the contrary, the 7d biomass yields are even lower for most consortia grown with 2% diesel (Fig. 3), while TPH degradation is almost complete under both settings. Please elaborate on this. Also, please rephrase the claim that “all cultured consortia [are] being able to assimilate diesel hydrocarbons” more carefully, because this does not seem to be true for some consortia at all.

Fig. 2: The error bars for the OD measurements are not shown or not visible. Please label all consortia on the x-axis.

Paragraph 3.2: Please rephrase the statement that “The use of NH4Cl as nitrogen source resulted in significantly greater bacterial growth and TPH degradation than the other inorganic sources” because i) “greater bacterial growth” is not a scientific phrase and ii) the difference of “greater growth” with NH4Cl does not seem to be significant between most of the nitrogen sources. Also, please rephrase “continue growth at all concentrations” because a continuation of growth is not shown here. Similarly, replace “degradation rates” because one-time measurements do not represent rates. Remove “trend”.

Fig. 4f): These results are somewhat problematic and probably show best, why OD measurements are not a proper way to estimate biomass increase in these kinds of experiments: as mentioned above one would expect a linear relationship between the amount of carbon source and the biomass yield, at least for a certain segment of the carbon source variation scale before substrate inhibition might play a role. However, these results show that nearly the same “final” OD is reached over a five-fold variation of the carbon source while the carbon source is almost completely removed at all concentrations. This needs to be discussed in detail.  

Paragraph 3.3: Please provide the ANOVA results. It would also help to follow the story if the tested categories (A-E) were named in the text. It does not become clear whether the RSM optimized conditions were also tested with consortium BS24. If so, please provide the results of this experiments.

Tab 1: Please provide the “+/- standard deviation” mentioned in the footnote.

Fig. 6+7: These figures should be combined.

Tab 6: Please add the name of the respective models to the first table column.

Discussion:

The discussion is generally well-written, however it is relatively short and superficial. A thorough discussion of the pitfalls and shortcomings (e.g. use of OD measurements, quality and quantity of the data used for the RSM analysis, …) of the study would be desirable. Other parts of the discussion, such as paragraph 5, don’t seem to be stringently connected to the presented study and should be rephrased.

Author Response

Comment 1

The manuscript by Roslee et al. describes an investigation into the diesel-degradation capabilities of microbial communities in Antarctic soil samples. After testing the efficiency of diesel degradation in different soil samples, the authors picked the most efficient sample and continued their analysis using one-factor-at-a-time and statistical response surface methodology analyses. This resulted in the identification of several factors that significantly influenced growth and diesel degradation efficiency. The study is in large parts well-conducted; however, some aspects need clarification before a potential publication in Biology. One major shortcoming of the study is that the authors do not analyze the taxonomic composition of their most promising consortium, which limits the applicability of this study even beyond the adverse geological and physicochemical conditions in the Antarctic mentioned by the authors.

Answer: Taxonomic identification through metagenomic approach will be carried out as an extension to the current study and the results are expected to be published in different publication.

Comment 2

Throughout the text: Please provide manuscript line numbers (e.g. use the manuscript template files you can find on the Biology Journal website).

Answer: line numbers have been added as requested

Comment 3

The introduction is well-written and provides sufficient background of the topic. However, it is very long. The authors could easily shorten the introduction by removing some details or moving them to the discussion, e.g. the detailed description of model fitness calculations, or of the use and shortcomings of OD measurements to estimate growth.

Answer: The correction has been made.

Comment 4

References:

Please use original research article only to support any scientific claims. E.g. reference 5 in the first paragraph does not seem to be a proper reference for the claim that the annual Diesel consumption amounts to 2 million liters annually. Please remove (or move) this reference and add a reference that actually provides data for this number. Similarly, references are missing in other instances, e.g. for the claims in the last sentence of the first paragraph.

Answer: 

1)The original source from Cripps and Shears (1997) was cited in place of Macoustra et al. (2015). Line 73

2)Added the publication to the reference list. Reference no 5.     Comment 5   Paragraph 2.1: Given that 20 g of soil sample were collected and later used in single aliquots in different experiments the phrase “consortium” used throughout the manuscript seems incorrect. If the soil samples were not homogenized before freezing one would expect many different “consortia” to occur in 20 g of soil, which would later be individually sampled for the following experiments. Please explain if the samples were well mixed before freezing or not.   Answer: 

1)Soil samples (20 g) were homogenised through vigorous shaking on vortex mixer for 15 min prior to sub-sampling (1 g). Line 165-166

2)The consortium used in subsequent testing came from the same initial soil that was maintained and sub-cultured weekly using the same growth conditions (NB as growth media, 10 °C and 150 rpm on rotary orbital shaker) prior to each experiment.     Comment 6   Paragraph 2.3: It does not become clear how the diesel degrading cultures were inoculated. What happened after adjusting the OD to 1.0? Were aliquots of this used to inoculate diesel-containing medium? What was the start OD? Were diesel containing cultures incubated statically or were they shaken? What type of incubation vessels were used?   Answer: 

1)Initially, the whole bacterial pellet obtained from 10 ml NB were used to prepare the standardised suspension (OD600 nm= 1). The OD600 values at 0 h of inoculation in BH medium supplemented with diesel using 1 ml suspension are uniform for every sample with values ranging from 0.02 to 0.03.

2)Refer supplementary data S3 for the growth curves of BS9, BS15a BS15b, BS23, BS24, and BS26

3)Please refer paragraph 2.5 for the growth settings

Comment 7

Paragraph 2.4: After describing the shortcomings of using OD to estimate growth in cultures with substrates that influence turbidity in the introduction, it seems counterintuitive that the authors chose this method for their analysis. At a minimum, the authors should describe how diesel influenced OD measurements throughout their incubations and discuss possible effects on the outcome of their analysis. It might also help readers who are not familiar with this kind of experiments to provide more details on the cultures, e.g. by describing whether the diesel was floating on top of the aqueous medium or whether it was dispersed.

Answer: 

1) Moved the sentence describing OD in the introduction to discussion section. Line 668-670

2) Added a few discussions on the possible effects of diesel for OD measurement and the its behaviour in liquid medium

Comment 8

The results are generally well-written, however the authors decided not to show selected results (e.g. 7 days growth curves) for some experiments. Since Biology requires that authors publish all experimental controls and make full datasets available, please provide these data in the supplement.

Answer: 

1)Refer Figure 2 for screening results of 28 soil samples

2)Refer Figure 3 for secondary screening results of six selected samples

3)Added supplementary data S1 and S2 to show how the selection was made and S3 to show the growth curves of the shortlisted consortia.

Comment 9

Paragraph 3.1: The authors describe in the Material and Methods section that they sampled the consortia-screening cultures every 24 h for OD measurement but do not show any data on this. Their results would be more convincing if these growth curves would be provided in a supplementary file and the reader could follow biomass increase in a time-dependent manner. This could also help to clarify the discrepancies in biomass yield between the first and second round of screening: One could roughly expect twice the biomass being produced when the amount of carbon source is doubled (1% vs. 2% diesel) given that the carbon source is quantitatively used up under both settings. However, this is not the case in the presented results. On the contrary, the 7d biomass yields are even lower for most consortia grown with 2% diesel (Fig. 3), while TPH degradation is almost complete under both settings. Please elaborate on this. Also, please rephrase the claim that “all cultured consortia [are] being able to assimilate diesel hydrocarbons” more carefully, because this does not seem to be true for some consortia at all.

Answer: 

1) Added supplementary data S1

2) The authors do not have decisive explanation to why the values of OD600 remain similar despites degradation for every diesel concentration below 2% were almost complete despite the.

3) Rephrased the claim that “all cultured consortia [are] being able to assimilate diesel hydrocarbons” by adding the sentence ‘except for BS17’ as the difference (BS17 vs control) was insignificant when compared to the control. However, all the other consortia were able to utilise at least a portion of the supplemented diesel. 

Comment 10

Fig. 2: The error bars for the OD measurements are not shown or not visible. Please label all consortia on the x-axis.

Answer: All of the consortia were labeled on the x-axis but there was an error in the figure dimension, causing some of the labels to be hidden. The error bars for OD were also included but happened to be too small to be shown in the figure

Comment 11

Paragraph 3.2: Please rephrase the statement that “The use of NH4Cl as nitrogen source resulted in significantly greater bacterial growth and TPH degradation than the other inorganic sources” because i) “greater bacterial growth” is not a scientific phrase and ii) the difference of “greater growth” with NH4Cl does not seem to be significant between most of the nitrogen sources. Also, please rephrase “continue growth at all concentrations” because a continuation of growth is not shown here. Similarly, replace “degradation rates” because one-time measurements do not represent rates. Remove “trend”.

Answer: 

1) Changed the sentence ‘greater bacterial growth’ to ‘increased microbial growth’. Line 328

2) The significant difference between levels/types within each factor were determined through one-way ANOVA followed by post-hoc Tukey test. The p-value of NH4Cl being <0.05 when compared to other inorganic N-sources concludes its significance.

3) Rephrased the sentence “continue growth at all concentrations” to simply ‘grow’ which refers to the ability of BS24 to propagate at distinct concentrations trialed although there was slight decrease in microbial growth beyond 0.5 g/L NH4Cl. Line 331.

Comment 12

Fig. 4f): These results are somewhat problematic and probably show best, why OD measurements are not a proper way to estimate biomass increase in these kinds of experiments: as mentioned above one would expect a linear relationship between the amount of carbon source and the biomass yield, at least for a certain segment of the carbon source variation scale before substrate inhibition might play a role. However, these results show that nearly the same “final” OD is reached over a five-fold variation of the carbon source while the carbon source is almost completely removed at all concentrations. This needs to be discussed in detail.  

Answer: 

1)There is yet any decisive answer to this since consortium behaviour is very hard to predict. The authors have tried CFU method but the varied growth rates (on agar plate) of many different organisms in the consortium makes it difficult for cell counting; while, dry cell weight method is restricted due to small working volume.

2)Diesel is a myriad of hydrocarbons (HC) typically C9-C18 that may become cytotoxic at high concentration. Not all of the HC, especially the heavier fractions were able to be degraded by microbes regardless of their concentration. 

Comment 13

Paragraph 3.3: Please provide the ANOVA results. It would also help to follow the story if the tested categories (A-E) were named in the text. It does not become clear whether the RSM optimized conditions were also tested with consortium BS24. If so, please provide the results of this experiments.

Answer:

1)See Table 3 for the Plackett-Burman ANOVA and Table 5 for the central-composite design ANOVA

2)Selected factors identity A, B, D, E were mentioned in-text See Table 1 for the factor identity (A-E). Line 405-406.

3)Only consortium BS24 (out of 28 samples initially) was optimised through RSM (consisting of PBD and CCD). Therefore, all the results from ODAT onwards are referring to BS24 only.

Comment 14

Tab 1: Please provide the “+/- standard deviation” mentioned in the footnote.

Answer: 1) added the standard deviation for each low/high level denoted by alphabets a-j. Line 247-248.

Comment 15

Fig. 6+7: These figures should be combined.

Answer: The figures have been combined

Comment 16

Tab 6: Please add the name of the respective models to the first table column.

Answer: Added the models name

Comment 17

Discussion:

The discussion is generally well-written, however it is relatively short and superficial. A thorough discussion of the pitfalls and shortcomings (e.g. use of OD measurements, quality and quantity of the data used for the RSM analysis, …) of the study would be desirable. Other parts of the discussion, such as paragraph 5, don’t seem to be stringently connected to the presented study and should be rephrased.

Answer: 

1) Discussed a few disadvantages of RSM. Line 640-648

2) Moved/Discussed the shortcomings of OD analysis for microbial growth assessment. Line 653-668.

Reviewer 2 Report

There is a typing error in the Fig.2, it says TPC degradation %, and it should be TPH. Please correct in the final version.

Author Response

Comment

There is a typing error in the Fig.2, it says TPC degradation %, and it should be TPH. Please correct in the final version.

Answer: Corrected . Figure 2

Reviewer 3 Report

In the present work, Fareez Ahmad Roslee and cols. report the characterization and optimization of a cold-adapted bacterial consortium (BS24), isolated from a soil obtained from the Antarctic Peninsula, to use diesel as the only carbon source. The effect of several factors on the metabolic capabilities of the consortium was studied to determine the optimal conditions for diesel degradation: temperature, salinity, N-source, N-source concentration, and pH. The authors used statistical response-surface methodology (RSM) to predict the optimal growth and diesel degradation conditions of the consortium.

The ms deals with a topic of high relevance and environmental interest that despite being studied for decades, represents a still unsolved problem for most Antarctic stations.  The experiments are well performed, and all the conclusions are supported by the presented results.

However, some minor details must be solved before the paper is accepted for publication.

Please include more details about the gravimetric method used for diesel quantification on cell cultures. The work is based on this methodology so is relevant to include more information and references explaining the method and validating their use in bacterial cultures. What is the sensitivity and accuracy of this method? there is a reference to validate it?.

Please indicate the percentage of diesel mineralized in controls, the abiotic degradation of diesel is interesting information that must be mentioned and discussed in the ms.

BS24 consortium was chosen for optimization. It is not clear why the authors decided to use this consortium considering the similar results presented for BS9, BS15a, and BS23. Maybe other criteria could be used to choose the better consortia to be optimized. The authors must clearly explain why BS24 was selected.

Fig 2.- where is BS24 in this graph? Why this consortium was not included?

Fig 2.- the abiotic control degrades approximately 10% of diesel in 7 days? How this is explained? there was any growth of cells (OD bear 0.1)? please explain this in the ms.

Fig.- 3 why some of these consortia were not included in Fig 2?

Fig.- 3 the control of Figure 2 shows similar TPH degradation but higher OD? why? please explain

Considering that the consortia selected in the study were obtained from Antarctic soils with the purpose of diesel degradation, the authors must indicate the characteristics of the soils obtained. There was evidence of anthropogenic contamination on these soils? Any evidence of diesel contamination? I suggest including in the ms the composition and characteristics of the soils used (at least soil BS24). Reporting the content of total hydrocarbons, N and C content, pH, and temperature will contribute to understanding the characteristics of the isolated consortia, to explain why some of them have better capabilities to metabolize diesel or growth at 10ºC. Also, I suggest determining the microbial composition of the soils and the BS24 consortia. This information will allow authors to understand the importance of specific bacterial species in the properties of the consortia, and also to predict potential metabolic interaction between species.

BS24 consortia can degrade diesel and use it as the only C or N source? What is the content of C and N in the Antarctic soils used? If this consortium will be used for in situ remediation of diesel in Antarctica, this can be relevant.

One of the most relevant contaminants associated with diesel are heavy metals, like Pb, Cd, and Cr. These metals are known to affect the capacity of soil microorganisms to metabolize PAHs. The content of heavy metals in the soil samples was determined?. Why did the authors decide not to evaluate the effect of heavy metals?

Author Response

Comment 1

Please include more details about the gravimetric method used for diesel quantification on cell cultures. The work is based on this methodology so is relevant to include more information and references explaining the method and validating their use in bacterial cultures. What is the sensitivity and accuracy of this method? there is a reference to validate it?.

Answer: 

1) Cited the reference used for gravimetric analysis https://doi.org/10.1007/s11270-008-9704-1 (Ref no 28). Line 147-148

2)The method is briefly explained in paragraph 2.4. Line 182-190

3) The method sensitivity is based on the type of weighing balance used. Analytical balance typically has uncertainty value of 0.0005 (4 d.p.). The gravimetric method is crude and prone to error if using highly volatile hydrocarbons, but quicker to analyse many samples per batch.

Comment 2

Please indicate the percentage of diesel mineralized in controls, the abiotic degradation of diesel is interesting information that must be mentioned and discussed in the ms.

Answer: 

1)Abiotic diesel loss in the control was explicitly shown only in the screening part (Figure 1 and Figure 2). However, the subsequent figures in OFAT and RSM analyses all take into account the abiotic loss of diesel using the expression provided in line 193.

2)Added a sentence pertinent to abiotic loss in the discussion. Line 671-674

Comment 3

BS24 consortium was chosen for optimization. It is not clear why the authors decided to use this consortium considering the similar results presented for BS9, BS15a, and BS23. Maybe other criteria could be used to choose the better consortia to be optimized. The authors must clearly explain why BS24 was selected.

Answer: 

In the secondary screening using 2% diesel, BS24 was observed to degrade diesel the highest while also maintaining growth as indicated through OD600 nm measurement (Figure 3). The selection of BS24 was supported by one-way ANOVA which revealed the growth of BS24 in 2% diesel is significantly higher compared to the other candidates although BS9, BS15a, and BS23 showed similar degradation percentage. Refer supplementary data S1 and S2.

Comment 4

Fig 2.- where is BS24 in this graph? Why this consortium was not included?

Answer: There was technical issue during the formatting of the manuscript. All 26 samples were included, it was just that the figure dimension was altered thus the labeling cannot be viewed although all the bars were there.

Comment 5

Fig 2.- the abiotic control degrades approximately 10% of diesel in 7 days? How this is explained? there was any growth of cells (OD bear 0.1)? please explain this in the ms.

Answer: 

)In Figure 2, approximately 17% diesel in the control was actually loss through abiotic factors (evaporation, aerosols formation during prolonged shaking). Therefore, the term ‘degradation’ is inaccurate with respect to experimental control since there was no bacteria involved. That’s why the control column bar was not included in the subsequent figures (but still taken into consideration while calculating for % TPH degradation)

2)Refer supplementary data S3 for the growth curve

3)Short explanation of abiotic loss was added in the discussion. Line 671-674.

Comment 6

Fig.- 3 why some of these consortia were not included in Fig 2?

Answer: Refer the answer to question number 6.

Comment 7

Fig.- 3 the control of Figure 2 shows similar TPH degradation but higher OD? why? please explain

Answer: There was error in the spreadsheet. The control is recalculated and Figure 3 is now updated.

Comment 8

Considering that the consortia selected in the study were obtained from Antarctic soils with the purpose of diesel degradation, the authors must indicate the characteristics of the soils obtained. There was evidence of anthropogenic contamination on these soils? Any evidence of diesel contamination? I suggest including in the ms the composition and characteristics of the soils used (at least soil BS24). Reporting the content of total hydrocarbons, N and C content, pH, and temperature will contribute to understanding the characteristics of the isolated consortia, to explain why some of them have better capabilities to metabolize diesel or growth at 10ºC. Also, I suggest determining the microbial composition of the soils and the BS24 consortia. This information will allow authors to understand the importance of specific bacterial species in the properties of the consortia, and also to predict potential metabolic interaction between species.

Answer:

1)There was a localised diesel spillage from one of the leaked underground pipelines connecting the SAB tank to the electrical generator of General Bernardo O’Higgins station (the collection point of soil BS24). However, the information on the volume spilt was not disclosed at the moment and the incident happened more than 3 years back. Since then, our research team have studied the area actively.

2)The BS24 soil characteristics (particle sizes, pH, salinity, and average temperature) were added in the results Paragraph 3.2; Whereas, the metagenomic microbial compositions are being studied currently; Hence the findings will be made available in different publication. Line 315-319

Comment 9

BS24 consortia can degrade diesel and use it as the only C or N source? What is the content of C and N in the Antarctic soils used? If this consortium will be used for in situ remediation of diesel in Antarctica, this can be relevant.

Answer: 

1)Diesel comprises of hydrocarbons (C and H) except for some species that might bear other atoms such as sulphur and nitrogen although at a very small percentage. Therefore, diesel act as the C source for the microbial consortia. The detailed N concentration in the original samples was not quantified but the amount of TPH was approximately around 0.66% or equivalent to 5.61 g per kg of soil as off January 2020.     Comment 10   One of the most relevant contaminants associated with diesel are heavy metals, like Pb, Cd, and Cr. These metals are known to affect the capacity of soil microorganisms to metabolize PAHs. The content of heavy metals in the soil samples was determined?. Why did the authors decide not to evaluate the effect of heavy metals?   Answer:

1)The levels of co-contaminants were not determined in this study. However, similar studies (which the authors partook) on the effects of various heavy metals using soil samples collected within the same area from where BS24 was obtained can be found below:

1)DOI: https://doi.org/10.24275/rmiq/Bio1072

2)DOI: 10.22438/JEB/41/5/MRN-1319

Reviewer 4 Report

The paper is well written and structured. No significant Engliesh revisions are required. In this paper, the authors investigated the potential of a microbial consortium from Antarctica for the diesel bioremediation. They used a double approach to test the biodegradation potential, and the statistical approach is really interesting, well suited and less improved in applications for polar microbes. Anyway, in my opinion the real lack of this paper is conceptual. We have not idea on the phylogenetic composition of the consortia, we don't know which are the bacterial taxa involved in the biodegradation of diesel. Indeed, we don't even know if it's an exclusively bacterial consortium. A characterization of the bacterial community would be necessary, since in the multi-part manuscript reference is made to hydrocarbon-degrading bacterial genera. So first of all you should refer to microorganisms, not only to bacteria. This means a re-contextualization of the entire paper, or an improvement of results, with evidences of bacterial characterization.

Moreover, there are some errors in the approach employed, or maybe I don't understand some steps. Why did you frozen your samples, to then isolate cultivable microorganisms? And why did you carried out the first step in NB? The most suitable method in this case is to process the samples immediately or within some hours by assessing microcosms.

See this paper, maybe it could be useful 

Microorganisms 2019, 7, 632; doi:10.3390/microorganisms7120632

I'm sorry, but I can't encourage the publication of the paper in this form. It needs conceptual adjustment in the research design and improvements of results. 

 Figure 2. Please correct TPC in the legend

Author Response

Comment 1

The paper is well written and structured. No significant English revisions are required. In this paper, the authors investigated the potential of a microbial consortium from Antarctica for the diesel bioremediation. They used a double approach to test the biodegradation potential, and the statistical approach is really interesting, well suited and less improved in applications for polar microbes. Anyway, in my opinion the real lack of this paper is conceptual. We have not idea on the phylogenetic composition of the consortia, we don't know which are the bacterial taxa involved in the biodegradation of diesel. Indeed, we don't even know if it's an exclusively bacterial consortium. A characterization of the bacterial community would be necessary, since in the multi-part manuscript reference is made to hydrocarbon-degrading bacterial genera. So first of all you should refer to microorganisms, not only to bacteria. This means a re-contextualization of the entire paper, or an improvement of results, with evidences of bacterial characterization.

Answer: 

1)Taxonomic identification through metagenomic approach will be carried out as an extension to the current study and the results are expected to be published in different publication.

2)The term ‘bacterial’ have been changed to ‘microbial’ throughout the manuscript to describe the uncharacterised consortium.

Comment 2

Moreover, there are some errors in the approach employed, or maybe I don't understand some steps. Why did you frozen your samples, to then isolate cultivable microorganisms? And why did you carried out the first step in NB? The most suitable method in this case is to process the samples immediately or within some hours by assessing microcosms.

Answer: 

1)The samples were collected in Antarctica but all the works mentioned in this manuscript were carried out in Malaysia, thus the soils need to be frozen during the storage and transport to preserve the microbial communities.

2)The microorganisms were then cultivated in NB so that they can be used in lab-scale diesel biodegradation studies using shake-flask (not a mesocosm study).

Comment 3

See this paper, maybe it could be useful Microorganisms 2019, 7, 632; doi:10.3390/microorganisms7120632

Answer: The authors have read the publication before. However, the study by Rizzo et al. (2019) used mesocosm approach to investigate the dynamics of microbial communities throughout the open-bioremediation process; Whereas, this study focused on the optimisation of cultivable microbial consortium in shake-flask set-up. The idea was to investigate the factors influencing diesel biodegradation together with their optimal levels so that they can be implemented in field study using controlled system (bioreactor-based clean-up).

Comment 4

Figure 2. Please correct TPC in the legend

Answer: Has been corrected

Round 2

Reviewer 1 Report

The authors did provide satisfying answers to most conserns of this reviewer. However some minor points should be adressed before publication of this manuscript.

Comment 1

The manuscript by Roslee et al. describes an investigation into the diesel-degradation capabilities of microbial communities in Antarctic soil samples. After testing the efficiency of diesel degradation in different soil samples, the authors picked the most efficient sample and continued their analysis using one-factor-at-a-time and statistical response surface methodology analyses. This resulted in the identification of several factors that significantly influenced growth and diesel degradation efficiency. The study is in large parts well-conducted; however, some aspects need clarification before a potential publication in Biology. One major shortcoming of the study is that the authors do not analyze the taxonomic composition of their most promising consortium, which limits the applicability of this study even beyond the adverse geological and physicochemical conditions in the Antarctic mentioned by the authors.

Answer: Taxonomic identification through metagenomic approach will be carried out as an extension to the current study and the results are expected to be published in different publication.

Reviewer reply: Of course, it is up to the authors to decide at what stage they want to try to publish their results. However, it is generally not recommended to split one correlative story into multiple single manuscripts. This manuscript in particular would benefit strongly from a taxonomic classification of the consortium, and, vice versa, a future manuscript describing the taxonomy of the consortium will lack significance due to a lack of interesting data about this consortium. Without a combination of both, the impact and the interest to the general readership of both manuscripts are significantly lower. Altogether, the decision whether the content of this manuscript in its current state is sufficient for publication in Biology lays in the hands of the editor.  

Comment 5   Paragraph 2.1: Given that 20 g of soil sample were collected and later used in single aliquots in different experiments the phrase “consortium” used throughout the manuscript seems incorrect. If the soil samples were not homogenized before freezing one would expect many different “consortia” to occur in 20 g of soil, which would later be individually sampled for the following experiments. Please explain if the samples were well mixed before freezing or not.  

Answer:  1) Soil samples (20 g) were homogenised through vigorous shaking on vortex mixer for 15 min prior to sub-sampling (1 g). Line 165-166

2)The consortium used in subsequent testing came from the same initial soil that was maintained and sub-cultured weekly using the same growth conditions (NB as growth media, 10 °C and 150 rpm on rotary orbital shaker) prior to each experiment.    

Reviewer reply: It is still not clear to me how these cultures were set up. Please try to describe this in a clear way so that everyone can follow. From your answer new questions arise: 1) Did you use 1 fresh g of your 20 g samples every time you started a new experiment or did you take 1 g of your sample once and sub-cultured the consortium from the respective NB culture on a weekly basis? 2) If you followed the first approach, did you also homogenize the whole 20 g sample before taking individual aliquots? 3) If you followed the second approach, a weekly transfer of a “microbial consortium” to rich NB medium is likely to favor growth of microorganisms that grow well under nutrient rich conditions and would i) change the taxonomic composition of the “consortium” significantly and constantly between the first transfer (high diversity) and later transfers (low diversity) and ii) would probably result in a pure culture of a single strain after a few transfers. Thus, the phrase “consortium” would be incorrect.. Again, taxonomic classifications of your cultures would solve this issue. Finally, the way this consortium has been maintained according to answer 2) is not described in the manuscript, please do so.

Comment 6   Paragraph 2.3: It does not become clear how the diesel degrading cultures were inoculated. What happened after adjusting the OD to 1.0? Were aliquots of this used to inoculate diesel-containing medium? What was the start OD? Were diesel containing cultures incubated statically or were they shaken? What type of incubation vessels were used?  

Answer: 

1)Initially, the whole bacterial pellet obtained from 10 ml NB were used to prepare the standardised suspension (OD600 nm= 1). The OD600 values at 0 h of inoculation in BH medium supplemented with diesel using 1 ml suspension are uniform for every sample with values ranging from 0.02 to 0.03.

2)Refer supplementary data S3 for the growth curves of BS9, BS15a BS15b, BS23, BS24, and BS26

Reviewer reply: Please cite supplemental figures and data in the manuscript.

3)Please refer paragraph 2.5 for the growth settings

Reviewer reply: Please describe growth settings also in paragraph 2.3

Comment 8

The results are generally well-written, however the authors decided not to show selected results (e.g. 7 days growth curves) for some experiments. Since Biology requires that authors publish all experimental controls and make full datasets available, please provide these data in the supplement.

Answer: 

1)Refer Figure 2 for screening results of 28 soil samples

2)Refer Figure 3 for secondary screening results of six selected samples

3)Added supplementary data S1 and S2 to show how the selection was made and S3 to show the growth curves of the shortlisted consortia.

Comment 9

Paragraph 3.1: The authors describe in the Material and Methods section that they sampled the consortia-screening cultures every 24 h for OD measurement but do not show any data on this. Their results would be more convincing if these growth curves would be provided in a supplementary file and the reader could follow biomass increase in a time-dependent manner. This could also help to clarify the discrepancies in biomass yield between the first and second round of screening: One could roughly expect twice the biomass being produced when the amount of carbon source is doubled (1% vs. 2% diesel) given that the carbon source is quantitatively used up under both settings. However, this is not the case in the presented results. On the contrary, the 7d biomass yields are even lower for most consortia grown with 2% diesel (Fig. 3), while TPH degradation is almost complete under both settings. Please elaborate on this. Also, please rephrase the claim that “all cultured consortia [are] being able to assimilate diesel hydrocarbons” more carefully, because this does not seem to be true for some consortia at all.

Answer: 

1) Added supplementary data S1

2) The authors do not have decisive explanation to why the values of OD600 remain similar despites degradation for every diesel concentration below 2% were almost complete despite the.

Reviewer reply: Please discuss possible explanations in the manuscript

3) Rephrased the claim that “all cultured consortia [are] being able to assimilate diesel hydrocarbons” by adding the sentence ‘except for BS17’ as the difference (BS17 vs control) was insignificant when compared to the control. However, all the other consortia were able to utilise at least a portion of the supplemented diesel. 

Comment 12

Fig. 4f): These results are somewhat problematic and probably show best, why OD measurements are not a proper way to estimate biomass increase in these kinds of experiments: as mentioned above one would expect a linear relationship between the amount of carbon source and the biomass yield, at least for a certain segment of the carbon source variation scale before substrate inhibition might play a role. However, these results show that nearly the same “final” OD is reached over a five-fold variation of the carbon source while the carbon source is almost completely removed at all concentrations. This needs to be discussed in detail.  

Answer: 

1)There is yet any decisive answer to this since consortium behaviour is very hard to predict. The authors have tried CFU method but the varied growth rates (on agar plate) of many different organisms in the consortium makes it difficult for cell counting; while, dry cell weight method is restricted due to small working volume.

2)Diesel is a myriad of hydrocarbons (HC) typically C9-C18 that may become cytotoxic at high concentration. Not all of the HC, especially the heavier fractions were able to be degraded by microbes regardless of their concentration. 

 Reviewer reply: Please discuss this in the manuscript

Author Response

Comment 1

Reviewer reply: Of course, it is up to the authors to decide at what stage they want to try to publish their results. However, it is generally not recommended to split one correlative story into multiple single manuscripts. This manuscript in particular would benefit strongly from a taxonomic classification of the consortium, and, vice versa, a future manuscript describing the taxonomy of the consortium will lack significance due to a lack of interesting data about this consortium. Without a combination of both, the impact and the interest to the general readership of both manuscripts are significantly lower. Altogether, the decision whether the content of this manuscript in its current state is sufficient for publication in Biology lays in the hands of the editor.  

Answer: Thank you for your comment. The authors share your concerns. However, this study as a whole comprises of many stages, which also includes fabrication/optimisation of bioreactor system and the metagenomic identification of microbial communities besides what have been presented in this manuscript. Therefore, if all the findings are to be published in a single manuscript, it will be a very lengthy one.

Comment 5

Reviewer reply: It is still not clear to me how these cultures were set up. Please try to describe this in a clear way so that everyone can follow. From your answer new questions arise: 1) Did you use 1 fresh g of your 20 g samples every time you started a new experiment or did you take 1 g of your sample once and sub-cultured the consortium from the respective NB culture on a weekly basis? 2) If you followed the first approach, did you also homogenize the whole 20 g sample before taking individual aliquots? 3) If you followed the second approach, a weekly transfer of a “microbial consortium” to rich NB medium is likely to favor growth of microorganisms that grow well under nutrient rich conditions and would i) change the taxonomic composition of the “consortium” significantly and constantly between the first transfer (high diversity) and later transfers (low diversity) and ii) would probably result in a pure culture of a single strain after a few transfers. Thus, the phrase “consortium” would be incorrect.. Again, taxonomic classifications of your cultures would solve this issue. Finally, the way this consortium has been maintained according to answer 2) is not described in the manuscript, please do so.

Answer: 

1) One gram of the soil sample was inoculated into 10 mL NB to produce the master culture. It was then subdivided into many seed cultures (100 μL) to ensure non-biased microbial composition for each experiment. The seed culture will be aliquoted into NB to increase the working size before it can be used for each set of experiment during the optimisation

2)Successive sub-culturing routine was done only to maintain the master culture so that the microbes remain metabolically active and serve as a backup if ever needed.

Comment 6

1) Reviewer reply: Please cite supplemental figures and data in the manuscript.

Answer: Cited in paragraph 3.1. Line 283-284

2) Reviewer reply: Please describe growth settings also in paragraph 2.3

Answer: 

The general growth medium was NB as described in paragraph 2.3, used to cultivate soil bacteria initially and for working volume amplification.

The medium used for diesel hydrocarbons degradation study was BH, which the default composition was described in paragraph 2.5

However, growing microbial consortia in both media followed the same initial settings (incubated on an orbital shaker at 10°C, 150 rpm for 2 d)

And paragraph 2.3 to 2.6 is a continuation. Therefore, mentioning the same steps will make them redundant

Comment 9

Reviewer reply: Please discuss possible explanations in the manuscript

Answer: Added a few sentences in the discussion part: Page 19-20: Line 674-683. “As mentioned elsewhere, diesel is a myriad of hydrocarbons in which some of them were known to interfere with membrane integrity and function. In Figure 4f, the authors reported an intriguing interaction between the effects of increasing hydrocarbons concentration towards the microbial growth. The similar growth peaks (p < 0.05) at day 7 for diesel concentrations of 0.5% to 2.5% may be attributed to a form of microbial adaptation strategy which prefers individual cell survivability instead of actively undergo cellular proliferation. At 3% and 4% of diesel concentrations, the microbial growth declined markedly as the cytotoxic effects become prominent and intolerable by majority of the microbial population. However, this claim is crudely speculative and therefore needs further investigation.”

Comment 12

Reviewer reply: Please discuss this in the manuscript

Answer: The same answer from comment 9. Page 19-20: Line 674-683.
